# Wastewater-Based Epidemiology to Describe the Evolution of SARS-CoV-2 in the South-East of Spain, and Application of Phylogenetic Analysis and a Machine Learning Approach

**DOI:** 10.3390/v15071499

**Published:** 2023-07-03

**Authors:** Jose A. Férez, Enric Cuevas-Ferrando, María Ayala-San Nicolás, Pedro J. Simón Andreu, Román López, Pilar Truchado, Gloria Sánchez, Ana Allende

**Affiliations:** 1Research Group on Microbiology and Quality of Fruit and Vegetables, CEBAS-CSIC, 30100 Murcia, Spain; joseantonioferez@gmail.com (J.A.F.); mayala@cebas.csic.es (M.A.-S.N.); ptruchado@cebas.csic.es (P.T.); aallende@cebas.csic.es (A.A.); 2Environmental Virology and Food Safety Lab (VISAFELab), Department of Preservation and Food Safety Technologies, Institute of Agrochemistry and Food Technology, IATA-CSIC, Av. Agustín Escardino 7, 46980 Valencia, Spain; enric.cuevas@iata.csic.es; 3Entidad Regional de Saneamiento y Depuración de Murcia (ESAMUR), Avda. Juan Carlos I, s/n. Ed. Torre Jemeca, 30009 Murcia, Spain; pedro.simon@esamur.com (P.J.S.A.); roman.lopez@esamur.com (R.L.)

**Keywords:** SARS-CoV-2, epidemiology, wastewater-based epidemiology, phylogenetic analysis, machine learning approach, molecular virology

## Abstract

The COVID-19 pandemic has posed a significant global threat, leading to several initiatives for its control and management. One such initiative involves wastewater-based epidemiology, which has gained attention for its potential to provide early warning of virus outbreaks and real-time information on its spread. In this study, wastewater samples from two wastewater treatment plants (WWTPs) located in the southeast of Spain (region of Murcia), namely Murcia, and Cartagena, were analyzed using RT-qPCR and high-throughput sequencing techniques to describe the evolution of SARS-CoV-2 in the South-East of Spain. Additionally, phylogenetic analysis and machine learning approaches were applied to develop a pre-screening tool for the identification of differences among the variant composition of different wastewater samples. The results confirmed that the levels of SARS-CoV-2 in these wastewater samples changed concerning the number of SARS-CoV-2 cases detected in the population, and variant occurrences were in line with clinical reported data. The sequence analyses helped to describe how the different SARS-CoV-2 variants have been replaced over time. Additionally, the phylogenetic analysis showed that samples obtained at close sampling times exhibited a higher similarity than those obtained more distantly in time. A second analysis using a machine learning approach based on the mutations found in the SARS-CoV-2 spike protein was also conducted. Hierarchical clustering (HC) was used as an efficient unsupervised approach for data analysis. Results indicated that samples obtained in October 2022 in Murcia and Cartagena were significantly different, which corresponded well with the different virus variants circulating in the two locations. The proposed methods in this study are adequate for comparing consensus sequence types of the SARS-CoV-2 sequences as a preliminary evaluation of potential changes in the variants that are circulating in a given population at a specific time point.

## 1. Introduction

On 11 March 2020, the WHO declared the current coronavirus disease (COVID-19) situation a global pandemic on the basis of “alarming levels of spread and severity, and by the alarming levels of inaction” [1,2]. This pandemic has caused a grave health crisis with serious consequences for the world economy due to the rigorous confinements being imposed, but it has also changed all the aspects in our lives, and science has not been an exception [3]. The struggle initiated in January 2020 to combat the severe acute respiratory syndrome coronavirus 2 (SARS-CoV-2) has become the top priority for all countries, which was translated into thousands and thousands of researchers putting all their energies into fighting this disease [4]. This situation has led to substantial investment in research funding that has, in turn, triggered a hitherto unprecedented volume of production of studies on SARS-CoV-2/COVID-19 [5]. It has been reported that investigation of COVID-19 became the most urgent priority; over 100,000 studies were published in 2020 alone [6,7].

The topics linked to SARS-CoV-2 are numerous, ranging from the epidemiology of the disease to treatments and impacts on different aspects of society. One of the main topics since the beginning of the COVID-19 pandemic has been wastewater-based epidemiology (WBE). Wastewater monitoring has been used as a successful, non-invasive, and early warning tool for monitoring the status and trend of COVID-19 infection and as an instrument for tuning public health responses [8]. The information obtained through WBE complements public health data, mostly because they provide evidence of the spread of the virus relative to a specific population in a given time. This could be illustrated by an increasing discrepancy between rising viral loads in wastewater, and confirmed cases may point to an undetected surge in infections [9]. This technique was very revealing at the beginning of the pandemic when public health data was scarce, and it can become very useful again in the present, when COVID-19 public surveillance seems to be drastically reduced. Currently, this environmental surveillance represents one of the main strategies implemented by many countries as a tool to help authorities to coordinate their exit strategies to gradually lift coronavirus lockdowns [10].

The usefulness of WBE is not only associated with the detection of the virus in the wastewater associated with a specific population, but also with an early warning alert for the detection of new variants of clinical concern for public health. Since March 2021, the European Commission adopted a recommendation on a common approach to establish and make greater use of systematic wastewater surveillance of SARS-CoV-2 as a new source of independent information on the spread of the virus and its variants in the European Union [10].

In the South-East of Spain, systematic surveillance of SARS-CoV-2 in wastewater started in March 2020 and continues to date, which resulted in a historical inventory of the SARS-CoV-2 concentration throughout the duration of the pandemic [8]. However, the WBE is not only considered an important method to trace viral circulation in a community in order to evaluate the prevalence, but it can also be used to determine the genomic diversity [11]. Massive sequencing techniques allow us to analyze a large number of SARS-CoV-2 genomes, including those present in symptomatic and asymptomatic persons [12]. The analysis of the sequences obtained from the RNA of wastewater samples allows the detection of the predominant, as well as low-frequency, variants, determining the variants that are circulating in a specific population. There are already many research papers demonstrating that whole genome sequencing (WGS) of SARS-CoV-2 provides additional data to complement routine diagnostic testing [9,12,13]. Data on the genetic diversity and emerging mutations in this pandemic have been crucial to understanding its evolution [14]. Phylogenetic trees and clustering analyses have been used to bring light to the international spread of SARS-CoV-2 and enabled the investigation of individual outbreaks and transmission chains in specific settings [15]. Previous attempts using genome-based phylogenetic analysis and machine learning approaches to analyze the evolution of SARS-CoV-2 have been already been published [16,17]. However, as far as we know, this approach has not been applied to data obtained from WBE in Spain.

The aim of the current study is to use wastewater monitoring and clinical data to provide an overview of the historical data for SARS-CoV-2 levels obtained in the South-East of Spain. This information will be combined with the results obtained using high-throughput sequencing techniques to describe the evolution of SARS-CoV-2. With the sequencing data provided by WGS, phylogenetic trees and machine learning approaches are used as a pre-screening for the identification of differences among the variant composition of different wastewater samples. The aim of this study is not to use phylogeny to describe the evolution of SARS-CoV-2.

## 2. Materials and Methods

### 2.1. Concentration Methods

Influent grab water samples were taken monthly from two wastewater treatment plants (WWTPs) located in the two main cities (Murcia and Cartagena) of the Region of Murcia (Spain) from March 2020 to March 2023. Concentration (200 mL) was performed using an aluminum-based adsorption–precipitation method, as previously described [8]. A final concentrate was then formed via centrifugation at 1900× *g* for 30 min, and the resulting pellet was resuspended in 1 mL of PBS, pH 7.4. 

Recovery controls were prepared by spiking PEDV (CV777 strain, kindly provided by Prof. Carvajal (University of Leon, Spain). For each sample, the percentage recovery was calculated by dividing the viral titer of concentrated sample by the titer of the recovery control. 

### 2.2. Viral Extraction, Detection and Quantification

Nucleic acid extraction of SARS-CoV-2 from wastewater concentrates was performed using an automated method with the Maxwell RSC Pure Food GMO and authentication kit (Promega) with slight modifications [18]. Firstly, 300 μL of concentrated samples were mixed with 400 μL of cetyltrimethylammonium bromide (CTAB) and 40 μL of proteinase K solution. The mixed sample was incubated at 60 °C for 10 min and centrifuged for 10 min at 16,000× *g*. Next, the resulting supernatant was transferred to the loading cartridge, and 300 μL of lysis buffer was added. The cartridge was then loaded in the Maxwell^®^ RSC Instrument (Promega, Madison, WI, USA) using the “Maxwell RSC Viral total Nucleic Acid” running program for the nucleic acid extraction. The obtained RNA was eluted in 100 μL nuclease-free water. Negative controls were included by using nuclease-free water instead of a concentrated sample. SARS-CoV-2 nucleic acid detection was performed via RT-qPCR using One Step PrimeScript^TM^ RT-PCR Kit (Perfect Real Time) (Takara, Kusatsu, Japan) targeting a genomic region of the nucleocapsid gene (N1 region) using primers, probes, and conditions previously described in CDC panel [19]. All RT-qPCR assays were performed in duplicate on a QuantStudio™ 5 Real-Time PCR (Applied Biosystems, Waltham, MA, USA). The Twist Synthetic SARS-CoV-2 RNA Control 1 (MN908947.3) and nuclease-free water were used as positive and negative controls, respectively.

### 2.3. Quantification of SARS-CoV-2 Variants

The prevalence of SARS-CoV-2 variants was assessed using five different duplex gene allelic discrimination TaqMan RT-qPCR procedures using primers, probes, and conditions previously described [20,21]. Alpha, Beta, Delta, Omicron BA.1, and Omicron BA.2 markers were mapped to the S gene (residues 69/70, 241/243, 157/158, 214, and 25/27, respectively); Gamma insertion is located between the end of ORF8 and the beginning of the N gene, and Omicron (B.1.1.529) deletion is mapped to the N gene (residues 31/33). The duplex RT-qPCR assay targeting 21765-21770DelTACATG mutation affecting residues 69/70 has been previously used to estimate the relative proportion of Alpha VOC, and is also suitable for estimating the proportions of Omicron BA.4 and BA.5 [21].

Each RT-qPCR analysis included duplicate wells with undiluted RNA and a 10-fold dilution to check for inhibition, as well as corresponding negative controls (amplification and extraction). Standard curves for genome quantitation of different variants were prepared using commercially available Twist Synthetic SARS-CoV-2 RNA Controls (Control 14, EPI_ISL_710528; control 16 (EPI_ISL_678597), control 17 (EPI_ISL_7926), and control 23 (EPI_ISL_15440143). The percentage of SARS-CoV-2 genomes containing each variant-specific mutation in the S gene was calculated using the formula: Variant% = GC/L (ProbeVariant)/[GC/L (ProbeVariant) + GC/L (ProbeNo_Variant)] × 100

### 2.4. SARS-CoV-2 Genome Sequencing and Analysis

Samples with RT-qPCR cycle threshold (Ct) values below 32-34 and a high recuperation percentage (≥25%), based on the recovery control (PEDV CV777 strain), were selected for sequencing analysis. Genomic sequencing of SARS-CoV-2 present in selected wastewater samples was carried out following the ARTIC protocol version 4 for retrotranscription using LunaScript^TM^ RT SuperMix (New England Biolabs, Ipswich, MA, USA) and amplification via multiplex PCR. Sequencing libraries were built using the Native Barcoding Kit (EXP-NBD-104 and EXP-NBD-114, Oxford Nanopore Technologies, Oxford, UK). The last purified product was eluted in 15 μL of elution buffer. Finally, the library was loaded on an R9.4.1 flow cell (FLO-MIN106) and placed onto a MinION Mk1C sequencer for a 36–48 h run.

After the sequencing runs, fast5 data files were base-called using Guppy (version 4.3.4, Oxford Nanopore Technologies, Oxford, UK) to generate fastq files (https://data-dataref.ifremer.fr/bioinfo/ifremer/obepine/lsem/data/dna-sequence-raw/ (accessed on 27 April 2023)). Successfully base-called reads were further analyzed following the ARTIC nCoV-2019 pipeline version 1.2.1.2 (ARTIC nCoV-2019 novel coronavirus bioinformatics protocol), which included demultiplexing, read filtering, primers, and barcode trimming. The resulting alignment file was used for single nucleotide variants (SNVs) calling using LoFreq version 2.1.5, with a minimum base quality of 20 and 20× coverage, relative to the Wuhan-Hu-1/2019 reference genome (GenBank: MN908947.3). Short indel calling was also performed using Lofreq after a preprocessing step to insert indel qualities. Samtools were used to read alignment files and to extract genome coverage percentages at different depths (10, 30 and 100). Samtools also allowed the extraction of mean genome coverage across the distinct amplicons trimmed for primer and overlapping sequences for each sample. For VOC analysis, we excluded samples with depth 30 coverage < 70%. Based on previous studies, single nucleotide variants (SNVs) and indels with coverage < 30, average quality < 30, frequency < 5%, and homopolymer run > 4 (for indels only) were excluded [22,23]. 

### 2.5. Clinical Data

Epidemiological data on COVID-19 in the Murcia Region were retrieved from the publicly available repository of the “Servicio de epidemiologia” of the “Consejería de Salud de la Región de Murcia” (http://www.murciasalud.es/principal.php (accessed on 17 March 2023)).

### 2.6. Phylogenetic Analysis 

Sixteen consensus SARS-CoV-2 sequences (in format .fasta) associated with the selected WWTPs (Murcia and Cartagena) from March to October 2022 were used for the phylogenetic analysis. These consensus sequences were generated using ARTIC bioinformatic pipeline for SARS-CoV-2 (ARTIC nCoV-2019 novel coronavirus bioinformatics protocol) specifically designed for Nanopore data.

The phylogenetic tree of the 16 consensus sequence types was generated using an alignment-free method with feature-frequency profile methodology [24].

The phylogenetic tree of the 16 consensus sequences was generated using an alignment-free method with feature-frequency profile methodology [24]. The proper performance of this methodology depends on the choice of the optimum value k for the sequence of length k denominated k-mer. The following formula was used to determine the optimum value for k [25]:k_(H_max) = log_4__*N*
where *N* is the length of the SARS-CoV-2 reference genome (GenBank: MN908947.3). In this study, the selected optimum k value corresponds to the positive integer value that verifies k > k_(H_max).

Then, *N* = 29.903 bp hence k_(H_max) = 7.43, which gives the value of optimum k = 8.

From the results obtained with the previous methodology based on 8-mer, the pairwise distance between each two genomes was estimated using the Jensen–Shannon divergence measure. As a result of this computation, we obtained a distance matrix that applied neighbor-joining (NJ) is a frequently used algorithm for constructing phylogenetic trees. The obtained distance matrix was used to generate the phylogenetic tree based on the NJ algorithm, to illustrate the relatedness between the consensus sequences. All analyses were conducted using R version 4.2.2. The specific R packages use for the analysis were:Phylogenic tree: seqinr, Biostrings, ape, textmineR.Clustering: openxlsx, dplyr, pheatmap, ggplot2.

### 2.7. Machine Learning Analysis 

The machine learning (ML) analysis was applied to the data corresponding to the 16 SARS-CoV-2 spike protein mutation profiles associated with the selected WWTPs (Murcia and Cartagena), covering the period from March to October 2022. The different WWTP profiles were classified by means of the Hierarchical Clustering ML technique [26]. In this case, the Agglomerative Hierarchical Clustering technique was implemented with the following options: Euclidean distance and Ward’s minimum variance method. All analyses were conducted using R version 4.2.2.

## 3. Results and Discussion

### 3.1. Overview of SARS-CoV-2 Genome Copies in Wastewater and Clinical Cases Detected in South-East of Spain

Since the beginning of the pandemic in March 2020, multiple waves of SARS-CoV-2 were recorded in Murcia (Spain). Three different situations can be identified between the WBE data and the clinical cases in the historical profile. The first of them (Figure 1A) is characterized by a low intensity, combined with an underrepresentation of the cases due to a low number of diagnostic tests performed. The second period is characterized by a high similarity between the fluctuation of the number of clinical cases and WBE data. This is due to the continuous performance of SARS-CoV-2 detection in wastewater and the intensive testing for SARS-CoV-2 in the population (Figure 1B). The third period of the historical profile is characterized by a similar outline regarding the WBE data, but since only the cases that are being hospitalized are still controlled by the public health authorities, the correlation is much lower (Figure 1C). However, based on the available data and knowing the high similarities observed in phase two of the historical profile, it could be assumed that transmission of the virus within this population is still high. Similar results have been observed by [27]. Therefore, identifying SARS-CoV-2 in WWTPs of these two cities from the southeast of Spain represents a very good proxy of the spread of the virus in the population. When comparing the WBE data with the clinically reported case counts, it is confirmed that the levels of SARS-CoV-2 in these wastewater samples changed in relation to the number of SARS-CoV-2 cases detected in the population, showing a very good correlation. Therefore, as previously reported, the use of WBE data has numerous advantages compared to clinical data, mostly because of a greater reliability (e.g., variations in access to testing and underreporting of asymptomatic cases) [8]. In addition, the low costs and easy implementation make WBE a key element in public health surveillance.

### 3.2. Evolution of the SARS-CoV-2 Variants in South-East of Spain

Numerous international efforts have arisen to identify SARS-CoV-2 in sewage systems during the pandemic. These efforts employ various analytical procedures that rely on RT-qPCR, genetic material preparation, and enrichment [28]. Throughout the pandemic’s progression, new variants of the virus have emerged, with specific mutations resulting in worldwide infections.

For all wastewater samples collected weekly from Cartagena and Murcia WWTPs, RT-qPCR was performed to identify and analyze the prevalence of UK/Alpha (Del 69/70), Beta (Del241/243), Delta (Del 157/158), Omicron (Del. 31/33), Omicron BA.1. (Ins 214), Omicron BA.2. (Del. 25/27, and Alpha/Omicron (Del. 69/70) variants of concern (Figure 2). 

RT-qPCR results show a 100% prevalence of the Omicron variant all through the studied period of 2022. Notably, between weeks 11 and 13 of 2022, a shift from the Omicron BA.1. variant to the Omicron BA.2. (presence of deletion 25/27 together with no presence of deletion 69/70) variant was observed. The 69/70 deletion, associated with the Alpha VOC at the first stages of the pandemic, was detected shortly after starting the studied period, as both Alpha and Omicron (BA.4 and BA.5) share this mutation. Thus, the concurrence of deletion 69/70 with deletion 25/27 from week 22 indicates a shift from Omicron BA.2 to Omicron BA.4 and BA.5. 

As other studies report, in mid-December 2021, the Omicron variant overtook the Delta variant, which had been the majority during the months of October and November 2021. The transition from Delta to Omicron occurred rapidly in 2-3 weeks in the month of December (weeks 49 to 51 of 2021) [29,30]; thus, our results, covering a posterior period of time with Omicron prevalence, are in line with the existing bibliography.

### 3.3. Phylogenetic Analysis of SARS-CoV-2 in Wastewater in South-East of Spain

The aim of this study is to apply phylogenetic analysis to compare the SARS-CoV-2 variants that are present in a population at a specific time point with other populations or other time points within the same population. The idea is to develop a pre-screening strategy to identify significant changes that can be derived from the introduction of a new variant. When a significant change is initially identified a deeper evaluation of the sequence analyses would be necessary. To compare different sequences, there are several approaches. The selection of one or another will depend on whether the genomes that will be compared, shared or not, have an alignable set of common genes. In cases where there is an alignable set of genes, two methods can be used, including: (1) multiple sequence alignment (MSA), where the identification a common subset of genes shared by all of the species is compared to build a MSA for each gene; and (2) the use of the gene profiling method, where the occurrence of each gene in a dictionary of genes is counted, forming a gene presence/absence profile [24]. On the other hand, if there is not an alignable set of common genes, the alignment-free method would be the only option. Following the method described by Sims et al. [24], a variation of the alignment-free method has been followed where the consensus sequence type can not only capture the distribution of each nucleotide, but also provide the covariance among nucleotides. Thus, global comparison of DNA sequences or genomes can be performed easily in the R application. 

As explained in the materials and method section, the NJ algorithm was used for constructing phylogenetic trees because of its fast operation and high accuracy, and is based on their similarity. In this case, the NJ algorithm was used to infer evolutionary relationships between different isolates obtained at different time intervals from the WWTPs of Murcia and Cartagena. The phylogenic tree was based on the complete genome sequences found in the wastewater at the different sampling times and locations (Cartagena and Murcia). In this case, the approach presented by Dong et al. [25], which represents each RNA sequence by a point in the R application, was used to investigate the variability of SARS-CoV-2 in different wastewater samples obtained from different locations and sampling times [24]. Mathematical algorithms were used to compare the whole genome of SARS-CoV-2 and to construct the tree-like diagram (dendogram), called a phylogenetic tree, which represents the evolutionary history of the sequences.

The aim was to determine if the consensus sequence types obtained from a specific sampling showed similarities or differences with other samples. Establishing a consensus sequence from wastewater samples to determine which is the SARS-CoV-2 variant affecting a population is not the most recommended methodology because sequences assembled from wastewater are a mix of many strains. However, in this study, the consensus sequence type was generated in a homogeneous way, which allowed them to be comparable. Each of the generated consensus sequence types is valuable as an overall view of all the SARS mutations that are present in a specific population at a specific time. It is the absolute comparison which gives relevant information regarding relevant changes over time in a specific population. Thus, this methodology could be used as a preliminary screening to determine if a drastic change in the variants that are present in a population occurs between one sampling time and another, or among different populations. If changes between two samples are observed, a more in-depth study could be needed to determine if this is due to the introduction of new variants. 

Similar approaches have been already used to determine if multiple lineages are present and circulating in a given population [9,31]. In this case, the selected algorithm-based phylogenetic methodology is a computational approach that has helped to identify the evolutionary relationships among these RNA sequences. 

The classification obtained based on the phylogenetic analysis shows that in most of the cases, samples obtained at close sampling times show a higher similarity than those samples more separated in time (Figure 3). Even though an outgroup is typically needed for a distance-based phylogenetic tree, in this study this was not essential for our purpose. For instance, the samples obtained in March and April 2022 from Murcia and Cartagena are placed together in the phylogenetic tree. The same happens for the samples obtained from Cartagena in May, June, July, and August 2022. However, in September and October 2022, samples between Murcia and Cartagena showed different consensus sequence types. Based on the variant information obtained for these two locations at these two sampling times, it could be observed that in the case of the WWTP of Murcia, the most abundant variant was BQ1.1, while in the case of Cartagena, the most abundant variant was BA.5, which could explain the differences observed in the consensus sequence types of the two locations. Other exceptions were the samples obtained in June and July 2022 in Murcia. In this case, these samples are placed separated from each other, mostly because different Omicron variants were detected. In the case of June, the most abundant Omicron variants were BA.1 and BA.2, while in July, the most abundant were BA.4 and BA.5. Differences were observed in the nucleotide composition, and consequently in the mutations.

### 3.4. Heat Map and Clustering Based on the Mutations of the SARS-CoV-2 Spike Protein Using Machine Learning Analysis 

A second analysis based on machine learning analysis was performed using the information obtained from the mutations found in the SARS-CoV-2 spike protein for each location and sampling times. Hierarchical clustering (HC) has been defined as an efficient unsupervised approach to unlabeled data analysis [26]. 

These data bring supplementary information on the similarities of the sequences that are circulating in a specific sample. Figure 4 shows the heat map linked to the mutations of the spike protein found for each sample. However, as the objective of the study was not to show an evolutionary description of SARS-CoV-2 in this Region of Spain, the evolutionary history for the sequences in the dendrogram have not been included. The algorithms used for this approach did not take information for every single nucleotide but from groups of 8 nucleotides. Therefore, the heat map shows the similarities among samples based on the nucleotides they share (Figure 4). It is relevant to highlight that del69/70 was almost close to 0 and del25/27 was only detected until May 2022. However, these two deletions were abundant from RT-qPCR, as indicated in Figure 2. One hypothesis is that the quality filters applied by the ARTIC protocol by default eliminated these deletions, making them invisible in the consensus sequence.

The heatmap represented in Figure 4 shows the probability (as a percentage) that each mutation in question is present in each sample. The gradation of colors allows the efficient representation of the percentages, so that the orange squares represent a 50% probability of the presence of the mutation. Based on the heat map, Figure 5 shows the clustering of samples based on mutations in the spike protein of SARS-CoV-2. This information provides valuable information about the genetic diversity of the virus and the emergence and spread of new variants. The spike protein is the primary target of neutralizing antibodies and plays a critical role in viral entry into host cells [32]. The colors included in Figure 5 indicate the different clusters (or similarity groups) formed by the different samples, based on the probability data (in percentage) that each mutation in question is present in each sample. 

It is important to understand that the clustering was based on the probability percentages of each of the mutations in the different WWTPs in the different sampled months, while the phylogenetic analysis was carried out based on the consensus sequences of each WWTP at all times. Therefore, the information provided by the clustering is based on a different concept than that used for the phylogenetic analysis. The information obtained in the two analyses can be considered complementary, but since they are based on different data, direct comparison is not correct. However, some general correlations can be found between the two analyses. Based on Figure 4, the samples obtained in March and April 2022 from Cartagena and Murcia show a high degree of similarity. Also, similar to that observed in the phylogenetic analysis, samples obtained in October 2022 in Murcia and Cartagena are very different, which corresponds well with the different variants that were circulating in the two locations. 

The methods proposed in this study seem to be adequate to compare consensus sequence types as a preliminary evaluation of potential changes in the variants that are circulating in a given population at a specific time point.

## Figures and Tables

**Figure 1 viruses-15-01499-f001:**
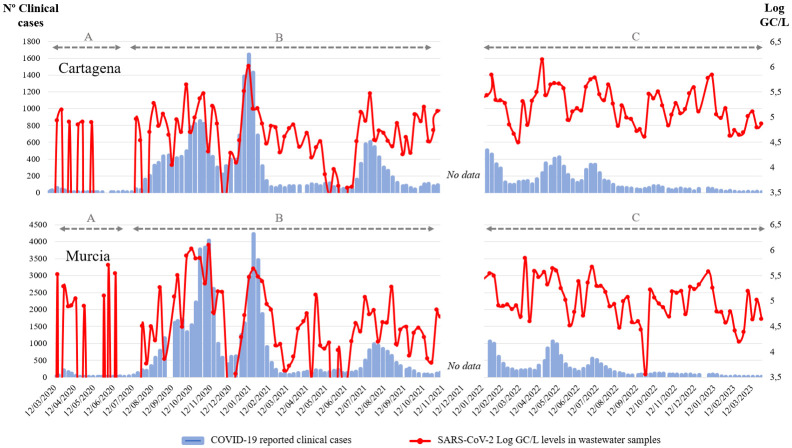
Representation of the number of reported clinical cases (blue bars) and the SARS-CoV-2 RNA levels (Log GC/L, red line) in wastewater samples collected from January 2020 to January 2023 in Cartagena and Murcia localities (Spain). “A” period covers the initial phase of the pandemic, when low number of diagnostic tests were being performed, leading to an underrepresentation of the cases; “B” period covers the main-stage of the pandemic, characterized by intensive testing for SARS-CoV-2 positives within the population; “C” period includes the final phase of the pandemic, when only cases from patients who were admitted to hospital were reported.

**Figure 2 viruses-15-01499-f002:**
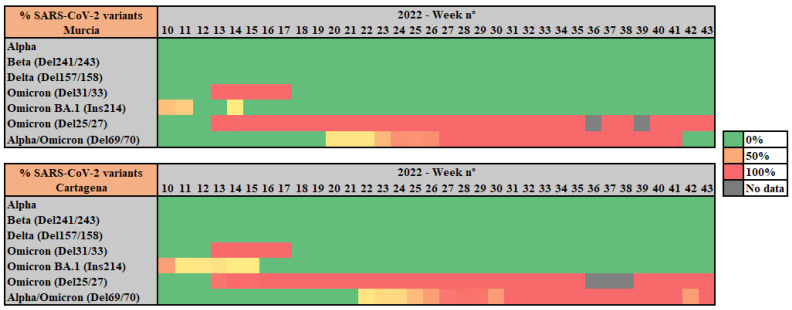
Heatmap of SARS-CoV-2 variants’ relative abundance in wastewater samples from Murcia and Cartagena WWTPs analyzed by duplex RT-qPCR.

**Figure 3 viruses-15-01499-f003:**
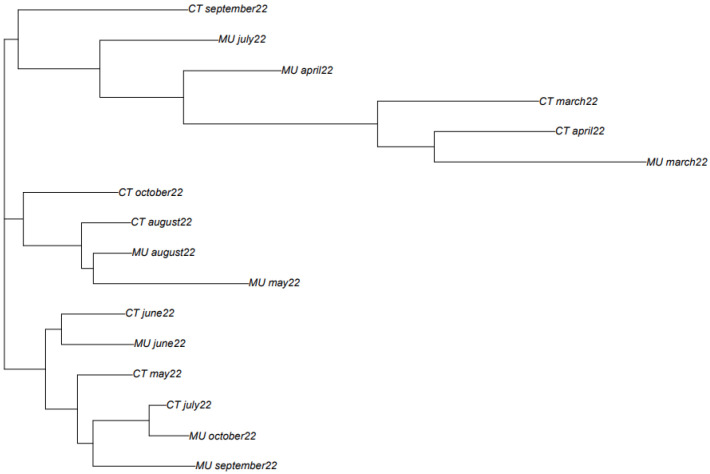
Phylogenetic tree created based on mathematical algorithms to compare the consensus sequence types of SARS-CoV-2 consensus sequence types found in the wastewater at different sampling times and locations.

**Figure 4 viruses-15-01499-f004:**
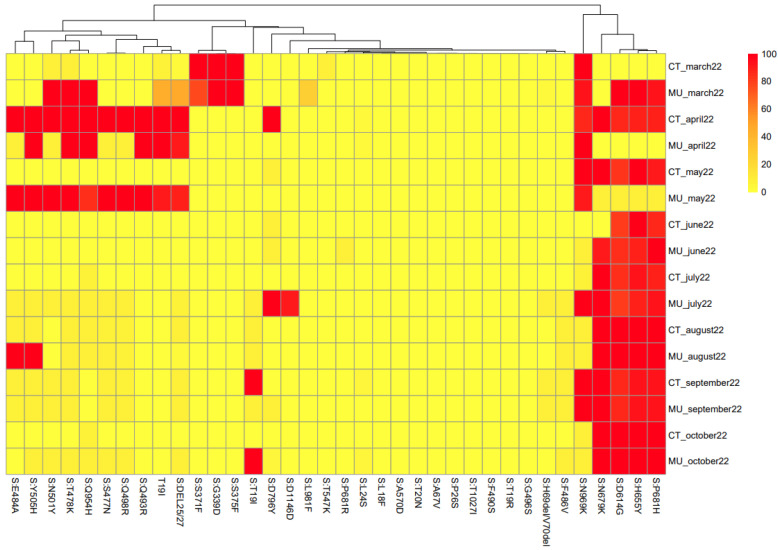
A heat map created based on the percentages of the mutations observed in the SARS-CoV-2 spike protein among different locations and sampling times.

**Figure 5 viruses-15-01499-f005:**
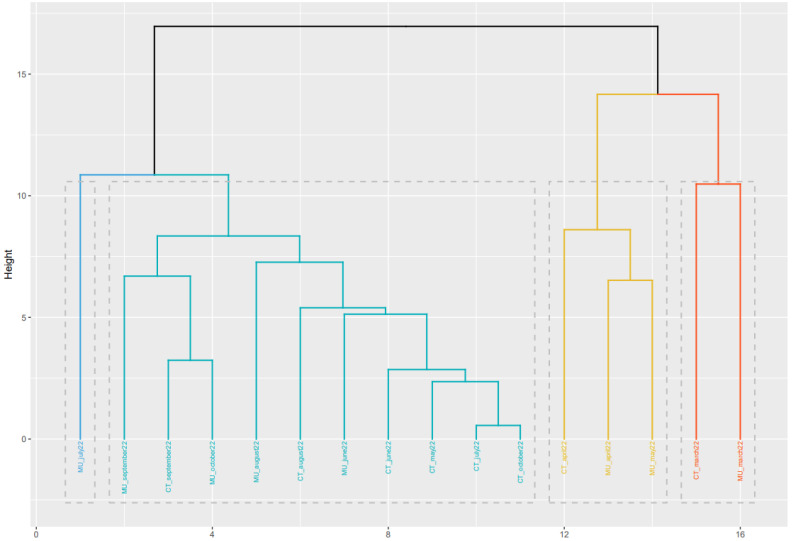
Cluster generated based on the heat map obtained from the percentages of the mutations observed in the SARS-CoV-2 spike protein.

## Data Availability

Not applicable.

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
