# Peer review of "Wastewater-Based Epidemiology to Describe the Evolution of SARS-CoV-2 in the South-East of Spain, and Application of Phylogenetic Analysis and a Machine Learning Approach"

_viruses, 2023, doi:10.3390/v15071499_

Round 1

Reviewer 1 Report

The authors of this paper provide an interesting example of workflow for a surveillance of SARS-COV-2 in waste water. 

The paper is clear and exaustive in every parts. In my opinion, it could be suitable for the publication.

Minor revisions:

In Material and Methods they should report the R packages.

The resolution of figures should be increased. 

Author Response

Please find enclosed the revised version of the manuscript submitted for publication. The paper has been amended following the reviewer's comments and a detailed response to reviewer’s concerns has been prepared. We would like to thank the reviewers for the valuable comments included which have helped us to improve the quality of the manuscript.

Reviewer 2 Report

The manuscript "Wastewater Based Epidemiology, Phylogenetic Analysis and Machine Learning Approach to Describe the Evolution of SARS-CoV-2 in the South-East of Spain" analysed the presence of SARS-CoV-2 in two Spanish WWTPs for almost 3 years, and illustrated the usage of phylogenetic analysis and machine learning method in the evaluation of SARS-CoV-2 evolution in wastewater. This study is impressive and the dataset is useful for other researchers. However, several methodological details are missing, and the presentation of results could be inaccurate, which need some major revisions.

Specific comments:

1.     There are some references cited incorrectly in the introduction. For example, in page 2, line 5, it stated “unprecedented volume of production of studies on SARS-CoV-2/COVID-19”, whereas the reference is only focus on SARS-CoV-2 vaccine. Also, in page 2, it stated “Previous attempts have been already published using genome-based phylogenetic analysis and machine learning approaches to analyze the evolution of SARS-CoV-2 (Li et al., 2020; Singh and Yi, 2021).”, but the two references seem not related to machine learning.

2.     In part 2.1, line 3, “extraction” should be “concentration”.

3.     Can the author add more details about the sampling information, such as volume, collection period, and frequency?

4.     According to cited reference (Randazzon 2020), it seems that the monitoring used different nucleic acid extraction methods in different periods (Nucleo Spin RNA kit vs Promega kit), also the control changed, will these changes affect the comparison of the amount of SARS-CoV-2?

5.     In part 2.3, “five different duplex gene allelic discrimination TaqMan RT-qPCR”, please specify which five. From the results 3.2, the RT-qPCR are seven (alpha, beta, delta, omicron, BA.1, BA.2 alpha/omicron).

6.     In part 2.4, “a high recuperation percentage (25%) were selected for sequencing analysis”, could the author explain how the recuperation percentage was calculated? Will this parameter affect the following NGS sequencing? If so, to what extent?

7.     In part 2.6, should “29.903” be “29,903”?

8.     In part 3.1, the three-year monitoring of SARS-CoV-2 in wastewater and then related it to clinical data is impressive. However, why the concentration of SARS-CoV-2 in wastewater starts from 4.5 (log10 GC/L)? It seems that the concentration in many weeks is below this number and therefore hidden in the figure.

9.     In part 3.2, there are many errors. “Delta (Del 157/188)” should be 157-158. “Omicron BA.1. (Ins 2/14)” should be Ins214, “Omicron (Del. 69/70)” should be “alpha/omicron”.

10.   In Figure 2, it is strange that the BA.2 variant could be detected from week 13 to week 43, since BA.2 was rapidly replaced by other omicron variants in Europe. For example, BA.4 and BA.5 also have del 25-27, the detection of this deletion does not mean it is BA.2, it could be other Omicron variants. Therefore, the figure 2 may not be accurate.

11.   In part 3.3, “we can further find an Accumulated Natural Vector (ANV) for each sequence”, could the author give more details in the method about what is the ANV, and how to find the ANV?

12.   In part 3.3, Is the “Algorithm-based phylogenetic methods” better than other common phylogenetic methods, such as NJ, UPGMA, ML, or Bayes? Or is this method more suitable for specific datasets?

13.   In Figure 3, usually an outgroup is typically needed for distance-based phylogenetic tree. Should the outgroup also need in your analysis? In addition in Figure 3, some results are difficult to understand, for example, the MU june and MU july are so distantly remote, could the author have some explanations?

14.   The sequence assembled from wastewater is a mix of many strains, will this kind of chimeric consensus sequence affect the phylogenetic analysis?

15.   In part 3.4, “Agglomerative Hierarchical Clustering (AHC) is the most adequate for the SARS-CoV-2 data (Li et al., 2022)”, the cited reference is nothing related to SARS-CoV-2. How is this statement concluded?

16.   In Figure 4, del69/70 is almost close to 0, and del25/27 could only be detected until May 2022. However, these two deletions were abundant from RT-qPCR (figure 2). Could the author explain why?

17.   Please improve the resolution for all figures. Especially Figure 5.

18.   It is better to have some paragraphs on the novel elements and the limitations of this study in the discussion.

19. Please check the format of the reference lists. It contains many different formats.

Author Response

(The authors gave the same response as above.)

Reviewer 3 Report

The manuscript provides data about SARS-CoV-2 detection from wastewater samples collected at two locations in Spain in 2022. The focus is on sequence analysis and the prediction of sequence variants. I think the study is important and of interest for the public. The manuscript is mostly well written, some minor spell check is required. I missed line numbering.

My major concern is that the manuscript appears unfinished. The title promises WBE, phylogeny and ML to describe the evolution of the virus. But I have not seen any real phylogeny that could be used to describe the evolution (i.e. changes over time compared to ancestors). I also completely miss the connection to clinical data. Which variants could be detected in infected persons during the period. Does this correlate with sewage data? What advantage do sewage data have over clinical data? If I want to make predictions using e.g. ML, I have to discuss the implementation and explain it carefully, because not everyone who might be interested is an expert in ML and/or phylogeny. Keyword discussion. This comes too short for me. As a reader, I see good data and am left alone with the interpretation. Overall, I like the approach and think the manuscript has potential. However, it should be clearly improved in the discussion. Some more specific points are listed below.

Translated with www.DeepL.com/Translator (free version)

Abstract

Line 4: use wastewater or sewage sample instead of water sample

Section 2.4: An Awk-script is mentioned. The script should be available for readers (supplements, Github etc.)

Sections 2.6 and 2.7: Did you use specific R packages for analyses? If, please mention and cite them.

Section 3.1.:

Please use ‘SARS-CoV-2 genome copies’ not ‘titer’ because data are based on RT-PCR, not titration of virus particles.

Line 4: rephrase ‘underestimation’ to e.g. ‘underrepresentation’ because it’s based on too few tests and not on estimations.

Line 7: not every SARS-CoV-2 positive test in the population might be due to Covid19. There are certainly also positive tests in persons without clinical outcome. Just say ‘intensive testing for SARS-CoV-2’

Is the correlation between clinical tests and WW test based on a statistical test (Figure 1 B and C)?

Section 3.2:

Figure 2: This figure shows no evolution of SARS-CoV-2. It is a heatmap with relative abundance of several tested SARS-CoV-2 variants.

The statement ‘Notably, between weeks 11 and 13 of 2022 a shift from Omicron BA.1. variant to Omicron BA.2. variant was observed’ needs to be explained. If we look at figure 2 in Cartagena  BA.2 present until week 15, in Murci only until week 11. Also, ‘RT-qPCR results show a 100% prevalence of the Omicron variant all along the stud-ied period of 2022.’ 100% would mean red colour? What about the orange (and one green) weeks 10 to 12. Was WT included in the testing?

Section 3.3:

R application is not precise. R is a programming language with many different packages available. It is not clear what exactly was (also not clear from material & methods). Please describe briefly to explain the readers what was done. I personally do not want to read every cited paper to understand the study.

Tree like diagram = dendrogram

What are the mathematical algorithms (see e.g. Figure 3 caption) you used? I do not see the evolutionary history for the sequences in the dendrogram. What is the branch length? Has the tree been rooted? It is kind of difficult to see history or ancestry without roots. Further, I assume figure 3 shows consensus sequences per sample? I understand the idea and its pretty nice to circumvent interpretation problems after using the Artic protocol. However, did you see an example that a ‘more depth study’ was needed to detect an emerging mutation (page 7)? That means did you observe single mutations that popped up and became significant over time? Is it furthermore possible to show the degree of deviation from the consensus? That would be interesting to see to estimate the potential of emerging variants.

Figures 4 and 5 need more explanation / description. What do we exactly see? Figure 4: does the heatmap show the relative abundance of different mutations per sample? I.e. present = red, absent = yellow, in between = ambiguities? That needs to be explained in the caption. Figure 5: the branch colours are? The dashed rectangles are? In addition, the labels are too small.

‘The information provided by the clustering is based on a different concept than that used for the phylogenetic analysis.’ Please elaborate.

Are the sequences publicly available?

Minor spell check is required. English Language is however fine.

Author Response

(The authors gave the same response as above.)

Round 2

Reviewer 2 Report

In Page 4, part 2.6, probably paragraph 2 forgot to remove. Otherwise, the concerns in the 1st version were addressed.

Author Response

thanks, we have corrected them in manuscript.